# The Covid-19 pandemic and the expansion of the mortality gap between the United States and its European peers

**Patrick Heuveline** *

Faculty Associate Director, California Center for Population Research at UCLA, Los Angeles, CA, United States of America

* heuveline@ucla.edu

**Data Availability Statement:** The minimal data set underlying the results described in the paper are already publicly available from The Human Mortality Database's Short-Term Mortality Fluctuation (STMF), https://www.mortality.org/

## Abstract

The mortality gap between the United States and other high-income nations substantially expanded during the first two decades of the 21$^{st}$ century. International comparisons of Covid-19 mortality suggest this gap might have grown during the Covid-19 pandemic. Applying population-weighted average mortality rates of the five largest West European countries to the US population reveals that this mortality gap increased the number of US deaths by 34.8% in 2021, causing 892,491 "excess deaths" that year. Controlling for population size, the annual number of excess deaths has nearly doubled between 2019 and 2021 (+84.9%). Diverging trends in Covid-19 mortality contributed to this increase in excess deaths, especially towards the end of 2021 as US vaccination rates plateaued at lower levels than in European countries. In 2021, the number of excess deaths involving Covid-19 in the United States reached 223,266 deaths, representing 25.0% of all excess deaths that year. However, 45.5% of the population-standardized increase in excess deaths between 2019 and 2021 is due to other causes of deaths. While the contribution of Covid-19 to excess mortality might be transient, divergent trends in mortality from other causes persistently separates the United States from West European countries. Excess mortality is particularly high between ages 15 and 64. In 2021, nearly half of all US deaths in this age range are excess deaths (48.0%).

## Introduction

The pre-pandemic expansion of the gap in mortality between the United States and other high-income nations is well documented [1–7]. Arguably the most salient measure of that mortality gap is the number of "excess deaths.". This number is estimated by subtracting from the actual number of deaths a counterfactual number of deaths obtained after replacing the prevailing sex- and age-specific mortality rates with more favorable rates. In recent months, the concept has been primarily used to assess the impact of the pandemic by substituting (lower) pre-pandemic rates for the actual rates. But the concept has long been used to assess

Data/STMF, the National Center for Health Statistics (NCHS), https://data.cdc.gov/NCHS/Provisional-COVID-19-Death-Counts-by-Sex-Age-and-S/9bhg-hcku, the United Nations Population Division, https://population.un.org/wpp/Download/Standard/Interpolated/, and the Johns Hopkins University Coronavirus Resource Center, https://coronavirus.jhu.edu.

**Funding:** The author benefited from facilities and resources provided by the California Center for Population Research at UCLA (CCPR), which receives core support (P2C-HD041022) from the Eunice Kennedy Shriver National Institute of Child Health and Human Development (NICHD). The funders had no role in study design, data collection and analysis, decision to publish, or preparation of the manuscript.

**Competing interests:** The author has declared that no competing interests exist.

the consequences of mortality differentials whether between population sub-groups, e.g., racial/ethnic differentials [8], or between populations [9].

A previous study tracked the annual number of excess deaths in the United States from 2000 to 2017 by estimating how many fewer deaths would have occurred had the country faced the same mortality rates as a composite of the five largest Western European countries (England and Wales, France, Germany, Italy, and Spain). With a combined population size very similar to that of the United States, these five countries provide a realistic benchmark for a large, diverse, and wealthy nation. The results were striking: the annual number of excess deaths estimated nearly doubled from 226,165 in 2000 to 400,732 in 2017, amounting to one in seven deaths that year (14.2%) [7]. This increase is partially driven by the growth of the US population. With the 2017 population, the 2000 number of excess deaths would have been substantially higher (297,922 deaths), but the population-standardized number of deaths did increase markedly (by +34.8% between 2000 and 2017). Moreover, this gap can be expected to have continued to expand during the pandemic, as provisional data suggested the United States experienced larger mortality declines than peer countries [10]. Another study estimated that the difference in Covid-19 mortality between the United States and the same five countries added 132,173 excess deaths in 2020, growing to 190,867 excess deaths for the 12-month period from April 1, 2020, to April 1, 2021 [11].

This study expands on these previous studies by extending the estimation of the annual number of US excess deaths relative to the same five European countries from 2017 to 2021, and of the specific contribution of differences in Covid-19 mortality from April 1, 2021, to the end of 2021.

## Material and methods

### Excess deaths from all causes, 2017 to 2021

This study first estimates excess deaths from all causes in the United States from the Human Mortality Database's Short-Term Mortality Fluctuation (*STMF*) [12]. These data include death counts and death rates for every week up to the end of 2021 in England & Wales, France, Germany, Italy, Spain, and the United States, by sex and five large age groups (0–14, 15–64, 65–74, 75–84 and 85+). Population-weighted average sex- and age-specific death rates for the five European countries (average European rates thereafter) were obtained by separately summing deaths and population exposure for each year, sex, and age group from each of the five countries, and by dividing the sum of deaths by the combined population exposure for each year, sex, and age group:

$$M_i^{Eu}(Y) = \frac{\sum_j D_i^j(Y)}{\sum_j P_i^j(Y)}$$

Where $M_i^{Eu}(Y)$ is the average European rate for sex and age group $i$ in year $Y$, $D_i^j(Y)$ is the number of deaths in sex and age group $i$ in country $j$ during year $Y$, and $P_i^j(Y)$ is the population exposure in sex and age group $i$ in country $j$ during year $Y$. As *STMF* data consist of weekly death counts and death rates, $D_i^j(Y)$ is first obtained by summing deaths across all weeks:

$$D_i^j(Y) = \sum_w D_i^j(w, Y)$$

where $D_i^j(w,Y)$ is the number of deaths in sex and age group $i$ in country $j$ during week $w$ of year $Y$, whereas $P_i^j(Y)$ is obtained by summing population exposure for each week of year $Y$, itself obtained by dividing the week's deaths by the week's death rate for that sex and age

group:

$$P_i^j(Y) = \sum_w \frac{D_i^j(w, Y)}{M_i^j(w, Y)}$$

where $M_i^j(w, Y)$ is the death rate in sex and age group $i$ in country $j$ during week $w$ of year $Y$.

Counterfactual numbers are derived by applying these average European rates to the US population exposure in the corresponding sex and age groups each year:

$$D_i^C(Y) = M_i^{Eu}(Y).P_i^{US}(Y)$$

where $D_i^C(Y)$ is the counterfactual number of US deaths in sex and age group $i$ during year $Y$ and $P_i^{US}(Y)$ is the US population exposure in sex and age group $i$ during year $Y$.

The number of excess deaths is then the difference between the actual and the counterfactual number of deaths in each year, sex- and age group:

$$D_i^X(Y) = D_i^{US}(Y) - D_i^C(Y)$$

where $D_i^X(Y)$ is the number of excess deaths in sex and age group $i$ in the United States during year $Y$ and $D_i^{US}(Y)$ is the actual number of US deaths in sex and age group $i$ during year $Y$. Annual numbers of excess deaths are then obtained by summing across all sex and age groups for the same year. Comparison with a prior estimate for 2017 suggest that having data only for five large age groups rather than for single year of age does not substantially modify the results (less than a 3-percent difference, see S1 File for details of the comparison).

To control for the contribution of population changes, another set of counterfactual numbers (standardized numbers thereafter) is derived by holding the US sex and age groups to their 2021 size:

$$D_i^{SC}(Y) = M_i^{Eu}(Y).P_i^{US}(2021)$$

where $D_i^{SC}(Y)$ is the standardized number of deaths in sex and age group $i$ in the United States during year $Y$ holding population exposure in sex and age group $i$ constant at its 2021 value.

Standardized numbers of excess deaths are then the difference between the actual and the standardized number of excess deaths for each year, sex, and age group:

$$D_i^{SX}(Y) = D_i^{US}(Y) - D_i^{SC}(Y)$$

where $D_i^{SX}(Y)$ is the standardized number of excess deaths in sex and age group $i$ in the United States during year $Y$.

### Excess deaths involving Covid-19, 2020 to 2021

To assess the specific contribution of Covid-19 mortality and its dynamics, counterfactual numbers of US deaths involving Covid-19 are estimated for each calendar year, 2020 and 2021, and for three twelve-month periods in-between (ending in the first, second and third quarter of 2021 respectively). Because countries report deaths involving Covid-19 using different age groupings [13], the above approach to estimate all-cause excess deaths was modified to rest instead on estimating a population-weighted Comparative Covid-19 Mortality Ratio [14] (CCMR) for the five European countries (average European CCMR thereafter). To compute this CCMR, the US sex- and age-specific rates of deaths involving Covid-19 for each period were first derived from Covid-19 death counts from the Centers for Disease Control and Prevention's National Center for Health Statistics (NCHS) [15] and mid-year US population

estimates by age and sex from the United Nations (*UN*) Population Division [16]:

$$^{C19}M_i^{US}(Y) = \frac{^{C19}D_i^{US}(Y)}{N_i^{US}(y)}$$

where $^{C19}M_i^{US}(Y)$ is the death rate for US deaths involving Covid-19 for sex and age group *I* in year *Y*, $^{C19}D_i^{US}(Y)$ is the *NCHS* number of US deaths involving Covid-19 in sex and age group *i* during year *Y*, and $N_i^{US}(y)$ is the *UN* estimate of the size of sex and age group *i* in the United States at time *y*, the mid-point of year *Y*.

These rates were then combined with the *UN* mid-year population estimates by age and sex to derive a counterfactual number of deaths involving Covid-19 in each of the five European countries and the United States and each year:

$$^{C19}D^{C,j}(Y) = \sum_i {}^{C19}M_i^{US}(Y) \cdot N_i^j(y)$$

where $^{C19}D^{C,j}(Y)$ is the counterfactual number of deaths involving Covid-19 in country *j* during year *Y* and $N_i^j(y)$ is the *UN* estimate of the size of sex and age group *i* in country *j* at time *y*.

Ratios of deaths involving Covid-19, as reported on the Johns Hopkins University (*JHU*) dashboard [17], to the corresponding counterfactual numbers above were calculated for each of the five European countries and the United States and each year:

$$CCMR^j(Y) = \frac{^{C19}D^{C,j}(Y)}{^{C19}D^j(Y)}$$

where $CCMR^j(Y)$ is the CCMR for country *j* in year *Y* and $^{C19}D^j(Y)$ is the *JHU* number of deaths involving Covid-19 in country *j* during year *Y*.

Due to slight differences between the number of US deaths involving COVID-19 reported by *NCHS* and *JHU*, the US CCMRs are not exactly equal to unity. To control for these reporting differences, the European CCMRs were divided by the US CCMR for the same year. The average European CCMR is then obtained as:

$$CCMR^{Eu}(Y) = \frac{\sum_j CCMR^j(Y).N^j(y)}{CCMR^{US}(Y).\sum_j N^j(y)}$$

where $CCMR^{Eu}(Y)$ is the average European CCMR for year *Y* and $N^j(y)$ is the *UN* estimate of the total population size of European country *j* at time *y*.

This ratio was taken to represent the fraction of US deaths involving Covid-19 that would have occurred in the absence of international differences in Covid-19 mortality and completeness of reporting deaths involving Covid-19. The number of US deaths involving Covid-19 that contribute to excess deaths is then:

$$^{C19}D^X(Y) = {}^{C19}D^{US}(Y) \cdot (1 - CCMR^{Eu}(Y))$$

where $^{C19}D^S(Y)$ is the counterfactual and $^{C19}D^{US}(Y)$ is the *NCHS* number of US deaths involving Covid-19 during year *Y*.

Finally, to estimate the contribution of Covid-19 mortality to the standardized number of excess deaths during the calendar year 2020, the *NCHS* number of US deaths involving Covid-19 in 2020 in the above equation was replaced by the standardized number of US deaths

involving Covid-19 in 2020 estimated as:

$$^{C19}D^{S,US}(2020) = \sum_i {}^{C19}M_i^{US}(2020) \cdot N_i^{US}(2021)$$

## Results

With the average European rates, the estimated number of US deaths for 2021 is 2,563,113 (Table 1). As there were 3,455,604 US deaths registered that year, the difference reveals 892,491 excess deaths in 2021. Holding US sex and age groups at their 2021 sizes shows that population changes only accounted for a small fraction of the 2017–2021 increase in excess deaths. The standardized number of excess deaths is 91.7% higher in 2021 than in 2017. In 2017, excess mortality increased the total number of deaths by 18.7%; in 2021, the increase in US deaths due to this excess mortality reached 34.8%.

Most of the increase in standardized number of excess deaths occurred between 2019 and 2021 (+84.9%, compared to +3.7% between 2017 and 2019). Differences in rates of deaths involving Covid-19 can thus be expected to explain the marked acceleration of the growing mortality gap between the United States and its European peers. Results shown in Table 1 suggest that more than half of the 2019–2021 increase in excess deaths (54.5%) can be attributed to international differences in Covid-19 mortality. The standardized number of US deaths involving Covid-19 that can be considered as excess deaths (relative to its European peer countries) increased from 136,594 in 2020 to 223,266 in 2021 (Table 1). This dynamic is further described in Fig 1 which shows the number of US deaths involving Covid-19, and the number excess deaths among those, and the ratio of the two for 2020, 2021, and three intermediate 12-month periods in-between. The number of US deaths involving Covid-19 that are excess deaths (sharded part of the bars) increased from 2020 to 2021 despite a modest decline in the total number of US deaths involving Covid-19 in a 12-month period (69,582 fewer deaths in 2021 than in the first 12 months of pandemic mortality, from April 1st, 2020, to April 1st, 2021). This modest decline was more than compensated by an increase in the fraction of excess deaths among those attributed to Covid-19. Relatively stable until mid-2021 (from 34.6% for 2020 to 37.2% for the mid-2020 to mid-2021 period), that fraction decreased to 29.2% in the 12-month period ending on October 1st, 2021, before bouncing back to 48.2% for the calendar year 2021.

Table 2 shows the distribution of 2021 excess deaths by sex and large age groups as well as the proportional increase in the number of deaths in each sex and age group that is contributed by excess deaths. In 2021, 31.0% of all excess deaths (both sexes combined) occurred to men between the ages of 15 and 64 and another 19.1% to women in the same age range. Altogether, more than half of the excess deaths were in that age range in 2021 (50.1%). Conversely, nearly half of all deaths between the ages of 15 and 64 are excess deaths, a higher proportion than in other age groups, so excess deaths increased the total number of deaths in this age group by 98.4% for women and 86.4% for men.

## Discussion

These results show that the marked increase in the standardized number of excess deaths in the United States relative to its peer West European countries between 2000 and 2017 (+34.8%) [7] was followed by an even larger increase, a near doubling in only four years (2017–2021). While death data by single year of age are not yet available to fully replicate the analysis from 2017 up to the present, data by large age groups provide a close approximation

**Table 1. Annual estimates of excess mortality in the United States (deaths from all causes, deaths involving Covid-19 and all other deaths), 2017 to 2021.**

|  | 2017 | 2018 | 2019 | 2020 | 2021 |
|---|---|---|---|---|---|
| US deaths, all causes |  |  |  |  |  |
| Annual number | 2,810,580 | 2,838,772 | 2,852,462 | 3,353,347 | 3,455,604 |
| Standardized number with 2021 population | 2,951,062 | 2,919,215 | 2,890,767 | 3,376,837 | 3,455,604 |
| Counterfactual standardized number with yearly European average rates | 2,485,445 | 2,468,694 | 2,408,098 | 2,624,849 | 2,563,113 |
| Standardized number of excess deaths | 465,617 | 450,521 | 482,668 | 751,988 | 892,491 |
| Percentage of counterfactual standardized number of excess deaths | 18.7% | 18.2% | 20.0% | 28.6% | 34.8% |
| US deaths involving Covid-19 |  |  |  |  |  |
| Annual number |  |  |  | 385,666 | 463,199 |
| Standardized number with 2021 population |  |  |  | 394,852 | 463,199 |
| Counterfactual standardized number with yearly European average CCMR |  |  |  | 258,258 | 239,933 |
| Standardized number of excess deaths involving Covid-19 |  |  |  | 136,594 | 223,266 |
| Percentage of standardized number of excess deaths from all causes |  |  |  | 18.2% | 25.0% |
| US deaths not involving Covid-19 |  |  |  |  |  |
| Standardized number of excess deaths not involving Covid-19 | 465,617 | 450,521 | 482,668 | 615,394 | 669,225 |

Notes: Author's calculations from the Human Mortality Database, Short-Term Fluctuation Series (STFS), with population-weighted average of European countries as counterfactual. Standardized numbers hold the size of the US population in each sex and age group constant at its 2021 value. European average rates (and CCMR) refer to the population-weighted average of the age- and sex-specific death rates (and CCMR) for each of the five European countries. The CCMR refers to the Comparative Covid-19 Mortality Ratio.

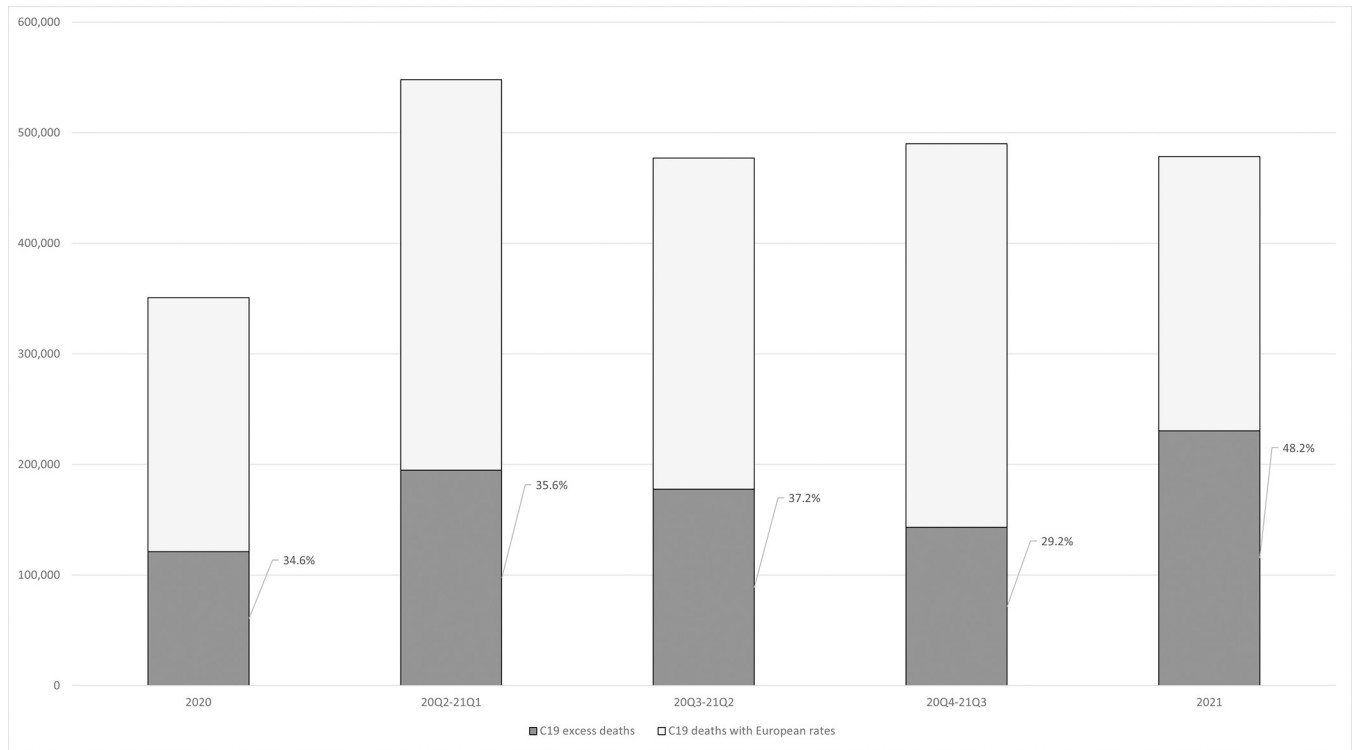

**Fig 1. Number of US deaths attributed to Covid-19 by 12-month period and proportion of Covid-19 deaths that are excess deaths (in percent).** Deaths attributed to Covid-19 are shown as the sum of Covid-19 deaths estimated by applying the population-weighted average of the European death rates from Covid-19 to the US population (light grey) and the excess deaths from Covid-19 (dark grey),.The first (left) and last (right) periods are calendar years. The three intermediate 12-month period are shifted by one quarter each: i.e., 20Q2-21Q1 refers to the period from April 1st, 2020, to April 1st, 2021.

**Table 2. Excess US deaths, and ratios to all US deaths, 2021, by sex and age group.**

|  | 0–14 | 15–64 | 65–74 | 75–84 | 85+ | All ages |
|---|---|---|---|---|---|---|
| Women | 5,685 | 170,268 | 112,059 | 94,141 | 53,944 | 436,097 |
| Men | 7,253 | 276,648 | 97,449 | 53,588 | 21,457 | 456,394 |
| Total | 12,938 | 446,916 | 209,508 | 147,729 | 75,401 | 892,491 |
| Distribution of all-age, both-sex excess deaths |  |  |  |  |  |  |
| Women | 0.6% | 19.1% | 12.6% | 10.5% | 6.0% | 48.9% |
| Men | 0.8% | 31.0% | 10.9% | 6.0% | 2.4% | 51.1% |
| Total | 1.4% | 50.1% | 23.5% | 16.6% | 8.4% | 100.0% |
| Percentage increase of number of deaths in sex and age-group |  |  |  |  |  |  |
| Women | 75.8% | 98.4% | 58.2% | 31.2% | 10.6% | 36.8% |
| Men | 78.6% | 86.4% | 30.4% | 14.2% | 6.1% | 33.1% |
| Total | 77.4% | 90.6% | 40.8% | 21.8% | 8.8% | 34.8% |

and demonstrate an unambiguous trend. From 2017 to 2019, the standardized number of excess deaths increased by another 3.7%, or 1.81% annually, a continuation of the 2000 to 2017 trend (+1.77%%). From 2019 to 2021, the standardized number of excess deaths increased 84.9% (+35.98% annually) to approach 900,000 in 2021 (892,491 deaths). The increase in standardized numbers of deaths due to this excess mortality also surged from 18.7% in 2017 (to 34.8% in 2021).

International differences in rates of death involving Covid-19 explain slightly more than half of the 2017–2021 increase in standardized numbers of excess deaths (+50.7%, 54.5% of the 2019–21 increase). These differences might be linked in part to differences in the prevalence of comorbidities associated with Covid-19 case-fatality rate. Through mid-2021, differences in Covid-19 death rates remained relatively stable [11]. For the second half of the year, the gap in Covid-19 death rates first narrowed as vaccines became available to the general population earlier in the United States than in Europe except for England and Wales. By the end of 2021, vaccination rates in the United States were plateauing at lower levels than those achieved in the West European countries [18], however, and the contribution of Covid-19 mortality to overall excess mortality grew larger in 2021 than it had been in previous 12-month periods since the beginning of the pandemic.

Perhaps more surprisingly, differences in mortality from other causes also continue to grow, contributing an additional 203,608 deaths to the standardized number of excess deaths between 2017 and 2021 (45.5% of the 2017–2021 standardized increase). Mortality from some causes might be expected to improve (e.g., from reduction in motor-vehicle traffic) [19] as well as to worsen for other causes (e.g., because of the impact of the pandemic on the quality of hospital care) [20] during the pandemic. While the net effect has been a substantial reduction in deaths from other causes in the five European countries, partially compensating the additional deaths directly attributable to Covid-19 [21], the United States has experienced an overall increase in deaths from other causes [22]. These might reflect in part a relatively higher degree of Covid-19 deaths being attributed to other causes [23], but rates of death from unintentional injuries have also been increasing in the United States [24], in particular those involving synthetic opioids [25] or alcohol [26]. While the contribution of Covid-19 should tilt excess mortality towards older ages, adult deaths between the ages of 15 and 64 thus continue provide the majority excess deaths (50.1%) and the share of deaths that are excess deaths is largest between these ages (49.6% for women, 46.4% for men).

The dramatic surge in excess mortality in the United States during the pandemic is, in Eileen Crimmins' words, "an acute demonstration of our chronic problem" [27]. So acute that

the number of US adult deaths was 90.6% higher that it would have been in the absence of "excess deaths" in 2021. Put differently 47.5% of US deaths of both sexes between ages 15 and 64 were due to this excess mortality, far exceeding the fraction of US deaths at ages 50 and over that were attributable to smoking in 2003 (24%) [28], or the fraction of adult deaths (ages 20–64) attributable to alcohol in Russia in 2002 (29%) [29]. Both these international differences and the pandemic's disproportionate impact on minority populations in the United States [8, 30] highlight social conditions as fundamental causes of diseases and deaths [31].

## Supporting information

**S1 File. Comparability with earlier excess-death estimates.**
(DOCX)

## Author Contributions

**Conceptualization:** Patrick Heuveline.

**Formal analysis:** Patrick Heuveline.

**Methodology:** Patrick Heuveline.

**Writing – original draft:** Patrick Heuveline.

**Writing – review & editing:** Patrick Heuveline.

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
