## [Decision Letter · Decision Letter 0]

25 Nov 2022

PONE-D-22-25276Another doubling of excess mortality in the United States relative to its European peers between 2017 and 2021PLOS ONE

Dear Dr. Heuveline,

Thank you for submitting your manuscript to PLOS ONE. After careful consideration, we feel that it has merit but does not fully meet PLOS ONE’s publication criteria as it currently stands. Therefore, we invite you to submit a revised version of the manuscript that addresses the points raised during the review process.

Based on the reviewers comments and my own reading, I believe the paper needs minor adjustments. One important issue: the paper needs to make clear the contribution in relation to yours PNAS (2022) paper. There are novel findings here, but it would be helpful to make it clear. The comments agree that the methodology could be revised to become more detailed and clear, some parts of it are not very clear and might be confusing. This leads to some difficult in reading the results.  Please, see also the detailed and careful comments by the reviewers. 

We look forward to receiving your revised manuscript.

Kind regards,

Bernardo Lanza Queiroz, Ph.D

Academic Editor

PLOS ONE

Journal Requirements:

2. Please change "female” or "male" to "woman” or "man" as appropriate, when used as a noun (see for instance https://apastyle.apa.org/style-grammar-guidelines/bias-free-language/gender).

 “No. The funders had no role in study design, data collection and analysis, decision to publish, or preparation of the manuscript.”

Reviewers' comments:

Reviewer's Responses to Questions

**Comments to the Author**

1. Is the manuscript technically sound, and do the data support the conclusions?

Reviewer #1: Yes

Reviewer #2: Yes

2. Has the statistical analysis been performed appropriately and rigorously? 

Reviewer #1: Yes

Reviewer #2: Yes

3. Have the authors made all data underlying the findings in their manuscript fully available?

Reviewer #1: Yes

Reviewer #2: Yes

4. Is the manuscript presented in an intelligible fashion and written in standard English?

Reviewer #1: Yes

Reviewer #2: Yes

5. Review Comments to the Author

Reviewer #1: Review of Another doubling of excess mortality in the United States relative to its European peers between 2017 and 2021

This is a short and interesting paper demonstrates the very concerning large increase in excess mortality in the US compared with the average of the 5 largest European countries in recent years. The analysis is mostly sound however I have a few queries about the choice of some of the measurements used,

The title of the paper and a major finding is that the annual number of excess deaths in the US versus the 5 European countries doubled between 2017 and 2021. This finding is based on the total number of excess deaths in the first row of Table 1 – which increased from 442,267 in 2017 to 892,491 in 2021. However, that measure includes the contribution of population change to the excess deaths of 42,317 (shown lower down in Table 1). Population change should be standardised for in analysis of trends in the number of excess deaths because it does not measure changes in population health but rather demographic trends (i.e. relative increase of the population). It would be more appropriate to compare the 442,267 figure in 2017 to 850,174 in 2021 that was standardised with the 2017 population – that is, the number of excess deaths did not double, but still increased substantially. The title and main findings of the paper should be adjusted to use the population-standardised excess mortality figures. Also, the “share of change in excess deaths” and other calculations in the manuscript should also be calculated based on population-standardised excess mortality.

The calculation of the excess deaths % is confusing. Conventionally, the excess mortality % is = (number deaths pop A – number deaths pop B) / number deaths pop B. However this paper divides excess deaths by the number of deaths in population A. It would be more insightful – and would also better demonstrate the large differences in mortality – if the former more conventional calculation were used. For example, the excess deaths in 2021 of 25.8% would instead be 25.8%/(100%-25.8%) = 34.8%, which is a more intuitive figure that shows that death rates in the US in 2021 were 35% higher than for 5 European countries. For ages 15-64 in Table 2, it would show that death rates in the US are almost double that in the 5 European countries.

Figure 1 would be more intuitive for the reader if simply the results were shown as a % as the quarterly results shown, rather than rolling annual figures presented.

The discussion of findings is quite limited and could point in the direction of future research to better understand why there has been a dramatic increase mortality in the US versus Western Europe – for example, measurement of the contribution of specific causes of death to the increase in excess mortality (in addition to Covid). It would also be good to describe some of the comorbidities of Covid mentioned in the 2nd paragraph of the Discussion, as well as mention differences in testing of Covid that might affect Covid death rates.

Some other comments:

- In the Abstract, it states “Applying average mortality rates…” – should this be “population-weighted average mortality rates…”?

- There should be a section title for the Introduction

- The first sentence “… is well documented” should have reference(s).

- Delete “so” from “the annual number of excess deaths estimated so nearly doubled from…”

- I had trouble finding the data for “Relatively stable until mid-2021 (from 34.6% for 2020 to 37.2% for the mid-2020 to mid-2021 period), that fraction decreased to 29.2% in the 12-month period ending on October 1st, 2021, before bouncing back to 48.2% for the calendar year 2021.”

- Suggest choosing another word other that “impressive” to describe the increase in excess deaths – e.g. “substantial”

Reviewer #2: This concise report provides estimates of excess deaths, defined as excess mortality of the U.S. relative to average of five W European countries, for the 2017-2021 period. The methodological approach is straightforward drawing from standard demographic analyses. A nice contribution is the decomposition of the excess deaths into those attributable to population composition change, COVID-19, and other-cause mortality. One main concern is that the report should be more explicit in the unique contribution of this analysis over the Heuveline (2022) PNAS paper. A few sentences in the Introduction would suffice. The methodology should also be made more explicit. Specific comments:

1. The gap in mortality between the U.S. and Europe has been long-standing, albeit growing. I don’t think “emergence” is the appropriate term as it implies that the gap emerged in the years leading up to the pandemic.

2. It might be useful to add a row for “Total Deaths” to Table 1 since percentage excess deaths are based on total recorded deaths.

3. The terms “ratio” and “share” are used in Table 1 – but percentages are shown in the Table. It may be helpful to use “Percent Excess Deaths” or “Percent Contribution…” as appropriate.

4. COVID-19 deaths: Were these measured by counting deaths with COVID-19 on the death certificate or calculations of excess deaths attributable to COVID-19? The method should be made clearer in this regard.

5. What is the explicand for the cause of death contributions? Is it the increase in excess deaths between each year and 2017? Is it the excess deaths for a given year? It is also not clear if the contributions are calculated with respect to the age-sex standardized excess deaths (or change in excess deaths).

6. PLOS authors have the option to publish the peer review history of their article (what does this mean?). If published, this will include your full peer review and any attached files.

Reviewer #1: **Yes: **Tim Adair

Reviewer #2: No

---

## [Author Response · Author response to Decision Letter 0]

5 Jan 2023

See rebuttal letter in separate file labeled "Response to Reviewers"

---

## [Decision Letter · Decision Letter 1]

3 Mar 2023

The Covid-19 pandemic and the expansion of the mortality gap between the United States and its European peers

PONE-D-22-25276R1

Dear Dr. Heuveline,

We’re pleased to inform you that your manuscript has been judged scientifically suitable for publication and will be formally accepted for publication once it meets all outstanding technical requirements.

Kind regards,

Bernardo Lanza Queiroz, Ph.D

Academic Editor

PLOS ONE

Additional Editor Comments (optional):

Reviewers' comments:

Reviewer's Responses to Questions

**Comments to the Author**

1. If the authors have adequately addressed your comments raised in a previous round of review and you feel that this manuscript is now acceptable for publication, you may indicate that here to bypass the “Comments to the Author” section, enter your conflict of interest statement in the “Confidential to Editor” section, and submit your "Accept" recommendation.

Reviewer #1: All comments have been addressed

2. Is the manuscript technically sound, and do the data support the conclusions?

Reviewer #1: Yes

3. Has the statistical analysis been performed appropriately and rigorously? 

Reviewer #1: Yes

4. Have the authors made all data underlying the findings in their manuscript fully available?

Reviewer #1: Yes

5. Is the manuscript presented in an intelligible fashion and written in standard English?

Reviewer #1: Yes

6. Review Comments to the Author

Reviewer #1: (No Response)

7. PLOS authors have the option to publish the peer review history of their article (what does this mean?). If published, this will include your full peer review and any attached files.

Reviewer #1: **Yes: **Tim Adair

---

## [Editor Report · Acceptance letter]

8 Mar 2023

PONE-D-22-25276R1 

The Covid-19 pandemic and the expansion of the mortality gap between the United States and its European peers 

Dear Dr. Heuveline:

I'm pleased to inform you that your manuscript has been deemed suitable for publication in PLOS ONE. Congratulations! Your manuscript is now with our production department. 

Kind regards, 

on behalf of

Dr. Bernardo Lanza Queiroz 

Academic Editor

PLOS ONE